# State Machine-Based Hybrid Position/Force Control Architecture for a Waste Management Mobile Robot with 5DOF Manipulator

Ionel-Alexandru Gal [1,*], Alexandra-Cătălina Ciocîrlan [1] and Mihai Mărgăritescu [2]

[1] Institute of Solid Mechanics of Romanian Academy, 010141 Bucharest, Romania; alexandra.ciocirlan@imsar.ro
[2] National Institute of Research and Development for Mechatronics and Measurement Technique—INCDMTM, 022404 Bucharest, Romania; mihai.margaritescu@gmail.com
* Correspondence: alexandru.gal@imsar.ro; Tel.: +40-741-575-553

**Featured Application: Robots with specific joint motors, with high potential for use in Industry 4.0 automated assembly lines.**

**Abstract:** When robots are built with state-driven motors, task-planning increases in complexity and difficulty. This type of actuator is difficult to control, because each type of control position/force requires different motor parameters. To solve this problem, we propose a state machine-driven hybrid position/force control architecture (SmHPFC). To achieve this, we take the classic hybrid position/force control method, while using only PID regulators, and add a state machine on top of it. In this way, the regulators will not help the control architecture, but the architecture will help the entire control system. The architecture acts both as a parameter update process and as a switching mechanism for the joints' decision S-matrix. The obtained control architecture was then applied to a 5DOF serial manipulator built with Festo motors. Using SmHPFC, the robot was then able to operate with position or force control depending on its designated task. Without the proposed architecture, the robot joint parameters would have to be updated using a more rigid approach; each time a new task begins with new parameters, the control type would have to be changed. Using the SmHPFC, the robot reference generation and task complexity is reduced to a much simpler one.

**Keywords:** hybrid control; state machine; Festo; PLC; friction force

## 1. Introduction

Waste management is a growing concern and problem [1] around the world and especially in the European Union. To help to solve this problem, rules, policies, and guidelines were presented to the public [2], as ways of lowering the amount of recyclable material going to waste dumps [3]. Communities are trying to recover more and more recyclable materials from day-to-day garbage, even when people are not using selective waste collection methods. For this, waste recycling companies usually hire people [4] to do the tedious work of selective recycling for different types of objects and materials (plastic and glass bottles, metal cans, etc.).

In order to aid companies when the workforce is scarce or to help employees when the work conditions are bad, automated processes have been developed to try and sort the waste [5] brought by garbage trucks. To this end, Diya et al. [6] proposed an intelligent system to help communities aim towards a greener environment. This goal can also be achieved through conventional selective methods using water for paper and plastic or magnets for ferrite-containing waste, while the remaining waste was compacted and incinerated or deposited in landfills. The other methods were to use video cameras on conveyors as Kokoulin et al. [7] proposed, or to use artificial intelligence as Sousa et al. [8] showed, to detect recyclable materials. Through different methods, using an intelligent

city platform created by Popa et al. [9], or by using a distributed architecture and machine learning built by Ziouzios et al. [10], recyclable waste can be moved to sorted containers, or even by combining both methods to maximize the selection efficiency. Both methods have their advantages and disadvantages [4,11], while using big and costly machinery.

An alternative to the automated conveyors and manual process of waste selective recycling is to use mobile robots that move inside a warehouse where the garbage is laid on the floor. While several robots conduct the search and recycle mission, stopping only to recharge their batteries, the process can continue practically non-stop without the need for human intervention, only requiring humans to replace the search material. This type of selective waste collection can reduce the need for human workers, therefore preventing the risks that arise from working with garbage and decomposing organic materials.

The proposed mobile robot has a mechanical arm/manipulator to grab the desired material and a video camera to detect recyclable objects. The serial mechanical manipulator has four positioning and orientation degrees of freedom and one degree of freedom for the gripper, resulting in a 5DOF mechanical arm for picking recyclable objects and placing them in a waste selective storage container.

The video camera detection method [7,8,12] is reliable and provides the position and orientation of the object with small detection errors that increase the sorting capability of recyclable material. Once the coordinates or the object is known, these are fed as a reference to the control algorithm. Having a certain robot architecture [13] and using specific Festo hardware [14], we created a hybrid position/force control method to achieve the best performance for this application. The main reason behind using the hybrid position/force control is to avoid the inverse matrix singularities of direct kinematics equations required when computing the end-effector reference values.

Starting with Raibert and Craig [15], the forefathers of the hybrid position/force control, the method has been developed [16] and used [17] in recent years to improve the interaction between robots and the environment [18], which is especially useful in our application. In 2019, Wang et al. [18] presented a hybrid position/force control used on a manipulator for close wall inspection, where the robot–environment interaction is treated as a mission consequence, and in 2020, Liu et al. [19] developed and used a hybrid position/force-controlled robot to drill and rivet metal plates with high precision ($\pm 0.1$ mm).

The hybrid position/force control (HPFC) methods [20] can and have been used in multiple areas of robot environments [21], combining them with new mathematical concepts [22] or fuzzy inference systems [23]. In 2011, Vladareanu and Smarandache [22] used neutrosophic logic to improve hybrid position/force control. In 2019, Ballesteros et al. [24] designed a second-order control feedback controller using the sliding mode control (SMC) [25] for an active orthotic mechanism, by using the system's dynamics and controlling each joint through a combined force and position reference, while gathering force sensor information. This approach uses the highly complex solution of designing a dynamic regulator while maintaining stability in a Lyapunov sense, while Rani et al. [26] developed an HPFC for task-coordinated manipulators using non-modeled system dynamics. Even if the dynamic equations were not used within the control diagram, they still ensured overall stability. Using the same approach, Peng et al. [27] created a hybrid control while using joint neural network observations on uncertain parameters and external perturbations which also uses dynamic equations and the Lyapunov stability to track and follow a path reference. Others have developed parallel manipulators with redundant actuation [28], some that learn the dynamics uncertainties to ensure stability [29], and even some hybrid control methods that use a fuzzy-neural network for multiple robot control [30].

When considering the type of regulator or control method to use, we took into account the sliding mode control (SMC) [25]. One reason was the experience the authors have in developing such control methods [25], and the other was that it has very good precision in compensating robot dynamics and external disturbances. As Zhang et al. [31],

Solanes et al. [32], and Ravandi et al. [33] have proven, the sliding mode control method can be combined with other methods to create different control algorithms depending on the required application. This approach may provide high accuracy where not needed, but it adds complexity to the mathematical model used in designing and developing the final control system. Because of this, the approach used in this paper was to avoid the high complexity of the model and focus on the application results, keeping the accuracy within the required parameters.

Different research papers have developed HPFC methods for specific applications when dynamic precision was required. Gao et al. [34] developed a 6DOF hydraulic manipulator that uses force and vision for positioning, while Han et al. [35] created a manipulator with a highly compliant end-effector. Some research handles the classic approach [18,19], while others combine the usefulness of the HPFC [20] with advantages of other methods [24–30] to increase stability and precision, to prevent perturbation or to compensate for uncertainties. These approaches, as reviewed in scientific papers [16,36,37], are necessary when the robot environment interaction requires high precision in positioning tasks or in order to combine position and force to safely interact with human patients or operators. Most of the published HPFC methods use a static approach in defining the joint matrix that separates the position-controlled joints from the force-controlled joints.

One way of preventing the static use of the S-matrix is to create a state feedback decision as Pasolli et al. [38] proposed. Their approach was to switch between position and force control on certain robot joints when the system required it, thus improving upon the classic hybrid position/force control of Raibert and Craig [15] by adding a decision layer to constantly change the position and force diagonal matrix S when certain events occur. Building on this method, we have proposed the hybrid position/force control combined with a state machine which will be called state machine-based hybrid position/force control (SmHPFC). This new method will update the parameters of the S-matrix as the need arises and depending on the task state of selective waste recycling.

The proposed method architecture was combined with a stick–slip analysis of the gripper's fingers to use a low force for gripping the detected recyclable waste objects, such as plastic bottles that can deform on a higher gripping force, or glass bottles that may slip with weaker force or break with a higher one. All of these were then implemented using Festo motors, Festo motor controllers, and Festo central control unit PLC.

## 2. System Description

The waste sorting robot is presented in Figure 1 [39,40] and Figure 2 [40,41], and is built from several separate systems that will work as a whole. The robot is the main objective of a national research grant [40] which aims to build a working prototype of the municipal waste sorting robot. Figure 1 presents one of several proposed 3D concepts, while Figure 2 presents the 3D design of the robot that has the following main components: (1) chassis, (2) housing, (3) drive wheels, (4) free wheels, (5) XYZ handling system, (6) gripper and (7) waste container [35]. The robot structure is already defined and published in a previous paper [41] and is not the subject of this paper.

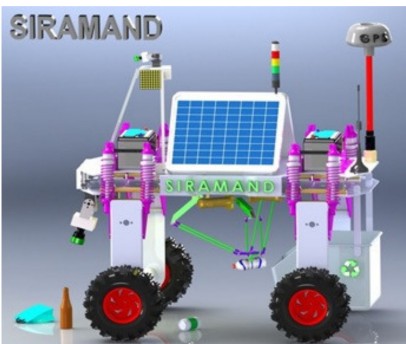

**Figure 1.** Initial concept of the waste sorting robot system [39,40].

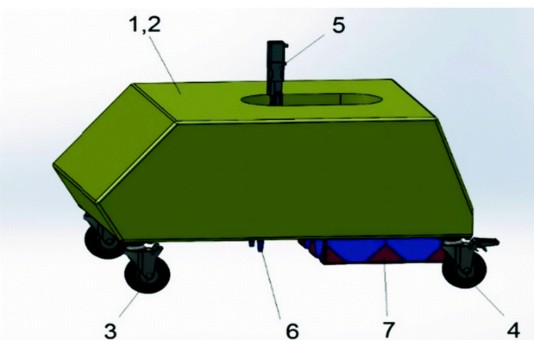

**Figure 2.** The mobile robot 3D model with the incorporated 5DOF manipulator [40,41].

An important component of the mobile robot is the drive wheels system for navigating through the waste, which was previously laid on the ground as a uniform thin layer. Another component that is not visible in Figure 2 is the vision system, which detects recyclable objects and sends the information to the 5DOF manipulator. The manipulator is the third important component of the mobile waste sorting robot, which grabs the detected recyclable materials and stores them in the attached waste container.

The gripping system is made of a five-degree-of-freedom serial robot, mounted on the mobile chassis and it was chosen as the simple approach in building the municipal waste sorting robots' manipulator. This is intended as a simple and easy solution for replacing the human workforce in searching for scattered recyclable materials such as plastic and glass bottles or aluminum cans, and it lowers the percentage of recyclable materials with a high decomposing time that reach a landfill.

The 5DOFs of the manipulator can be separated into three categories: positioning, orientation, and interaction. The first set, required for positioning the end-effector, is composed of three linear motors. These 3DOFs move the gripper on all three axes (OX, OY and OZ), achieving a positioning system within the robot manipulators' confined workspace. In our robotic system, the horizontal XOY movement is ensured by a Festo planar surface gantry EXCM, while the OZ vertical displacement is given by a Festo electrical slide EGSK. The orientation system uses a one-degree-of-freedom Festo rotary motor ERMO to rotate the end-effector around the vertical axis OZ to align the gripper with the object/bottle. Once the position and orientation are set, the last degree of freedom is the end-effector which, in our case, is a Festo HGPLE gripper that uses a dual linear motor that moves two rigid bodies (the gripper jaws) in two opposite directions on a single axis. The first direction, which is considered to be the positive one, is when the two jaws of the gripper are closing into each other and the second is when both move away from each other in the negative direction.

To control the serial robot for selecting the recyclable materials, we chose the hybrid position/force control that allows us to position and orient the gripper within the workspace, through a position-based control law, and to grip the object using a force-based approach. Moreover, the gripper can be controlled in position for opening the jaws or in force for gripping the objects. The control type of the gripper has to switch in real-time, depending on the state of the robot and the mission objective. This is why a hybrid position/force control law with a state-machine decision system was chosen as the control algorithm that drives the 5DOF manipulator.

## 3. Decision Algorithm

The decision algorithm is the component that turns the static hybrid position/force control method into a dynamic control by changing the Raibert and Craig [15] static configuration of the S-matrix, and updating its values depending on a deterministic state machine. The diagram of the overall state machine is presented in Figure 3. Here, one can see multiple system states followed by one or more transitions. Table 1 presents the state description and Table 2 presents the description of the transition.

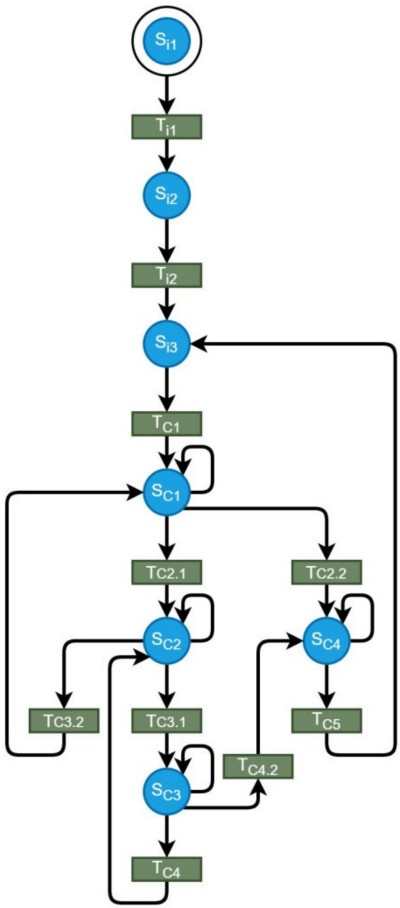

**Figure 3.** The main 5DOF robot state machine for waste selection.

**Table 1.** State machine states list and descriptions.

| State | Diagram | Description |
|-------|---------|-------------|
| $S_{i1}$ | Main | The initial state of the entire system. |
| $S_{i2}$ | Main/Homing | Initialization complete. |
| $S_{i2.1}$ | Homing | Homing process started. |
| $S_{i3}$ | All | Homing done or motion complete. |
| $S_{C1}$ | Main | Positioning in XOY plane and rotate gripper. |
| $S_{C2}$ | Main/Force Control | Positioning on OZ axis (move gripper up or down) and rotate around OZ for gripper orientation. |
| $S_{C3}$ | Main/Force Control | Ready to start force control (grip object). |
| $S_{C3.1}$ | Force Control | Doing force control for selected reference. |
| $S_{C3.2}$ | Force Control | Gripping force is stable while holding the object. |
| $S_{C4}$ | Main/Position Control/Force Control | Stop force control (done). Switching to position control and open gripper. |
| $S_{C5}$ | Position Control | Doing position control for selected reference. |
| $S_{ES}$ | Position Control/Force Control | Emergency stop. |

To keep the state machine clear and easy to understand, several components were detailed in additional diagrams presented in Figure 4a–c. These are the homing state

machine, the position control state machine, and the force control state machine. Their state and transition descriptions can be also found in Table 1.

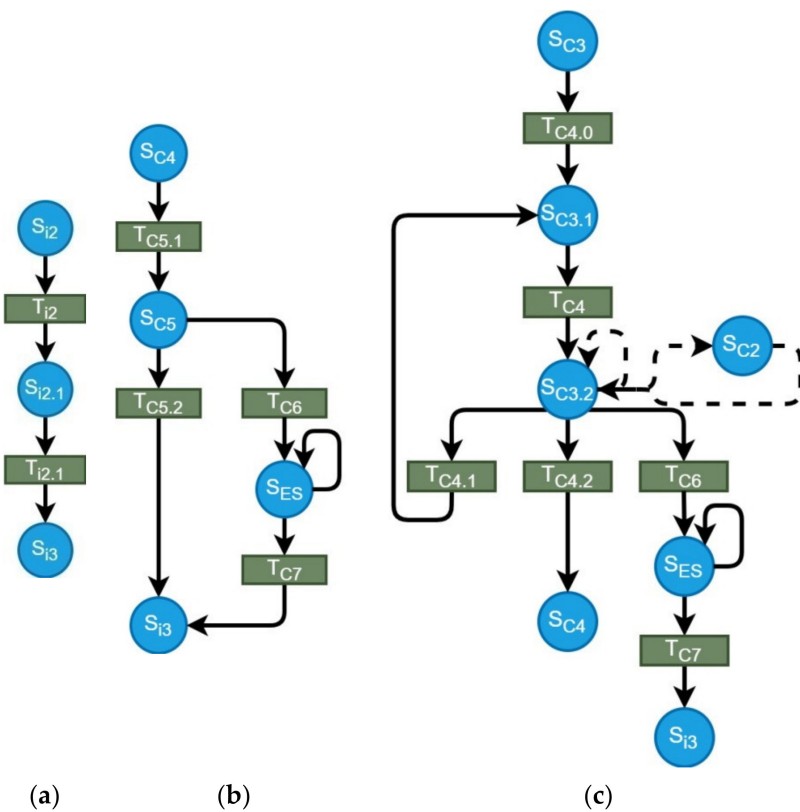

**Figure 4.** State machine sub-sections: (**a**) the homing state machine; (**b**) the position control state machine; (**c**) the force control state machine.

The homing state machine is very simple. The homing process is started sequentially on all five DOFs (state $S_{i2.1}$). When the homing process is complete, the 5DOF system transitions to the stable state $S_{i3}$. This state is the starting point for all following control sequences. When a new task is received (transition $T_{C1}$), the system begins to complete it, starting with the first two translation joints for positioning on XOY plane. Then, the vertical motion and orientation begin ($S_{C2}$) and on its completion, the force control takes over for the gripper joint by changing the control type for this degree of freedom within the S-matrix. Thus, by using force control, the gripper attempts to grab the target object using the reference force given as input to the control system. Figure 4c details how the force control handles different events. While the gripper force holds, the 4DOF positioning system lifts the object and position the gripper above the recycling tray. At this moment, the gripper force control ends by updating the S-matrix, and the object is dropped by opening the gripper's jaws using position control. After the object is dropped into the recycling tray, the task is complete, and the 5DOF system can start a new task.

To better explain the sequence, the decision Algorithm 1 that can be more easily understood is outlined below.

---

**Algorithm 1** Recycling process using the hybrid position/force control

---

for (i ∈ [1; 5])
　　Init ($M_i$)
　　Homing ($M_i$)
while (!ES &&New Task)
　　get ($Obj_{Type}$, $T_{xyz}$, $R_z$)
　　if ($\Delta P + z > \varepsilon_{Pz}$)
　　GripperMove ($T_{top}$)
　　if (!$G_{open}$)
　　Jaws (open)
Compute ($F_{jaws}$)
GripperMove ($T_{xy}$)
GripperRotate ($R_z$)
GripperMove ($T_z$)
Switch ($To_{ForceControl}$)
　　Update ($S_P$, $S_F$)
Jaws (close)
while ($\Delta F_{jaws} \leq \varepsilon_{Fjaws}$)
GripperMove ($T_{top}$)
GripperRotate ($R_{tray}$)
GripperMove ($T_{tray}$)
Switch ($To_{PositionControl}$)
　　Update ($S_P$, $S_F$)
Jaws (open)

---

where:
　　$M_i$ = motor i,
　　ES = Emergency Stop,
　　$Obj_{Type}$ = Object type (metal, glass, plastic),
　　$T_{xyz}$ = target coordinates vector,
　　$T_{top}$ = coordinates of top gripper position,
　　$T_{tray}$ = coordinates of waste tray position,
　　$R_z$ = target rotation on the Z-axis,
　　$R_{tray}$ = waste tray rotation on the Z-axis,
　　$F_{jaws}$ = gripper reference force required to grab the current target,
　　$\Delta P_z$ = position error on the Z-axis,
　　$\varepsilon_{Pz}$ = maximum allowed positioning error on the Z-axis,
　　$\Delta F_{jaws}$ = gripper jaws force error,
　　$\varepsilon_{Fjaws}$ = maximum allows gripper jaws force error,
　　$S_P$ = position matrix joints,
　　$S_F$ = force matrix joints.

## 4. Influence of Friction Force Analysis on the Gripper's Control Method

To pick up recyclable objects from the municipal waste collection, the gripper needs at least two fingers. For this, we created a 3D finger design which is presented in Figure 5a–c. These were designed as simple fingers that can grab different object types and shapes. From the 3D design, the fingers were 3D printed using ABS material and a Zortrax M200 3D printer.

When complete, the fingers were attached to the linear motor, resulting in a two-finger gripper mechanism presented in Figure 6. One can observe in Figure 6 a small deformation of the ABS fingers during the gripping action. This ABS finger's flexibility and deformability ensure a slower transmission of the force towards the actuation of the gripper. Certain protection of the electric actuation of the gripper is thus ensured.

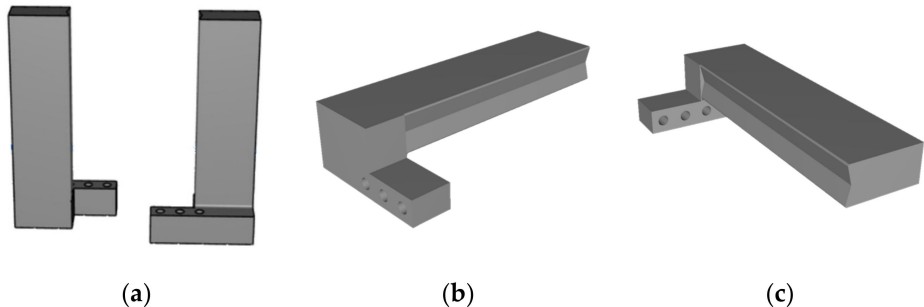

**Figure 5.** Gripper 3D fingers: (**a**) front view; (**b**) bottom side view; (**c**) top side view.

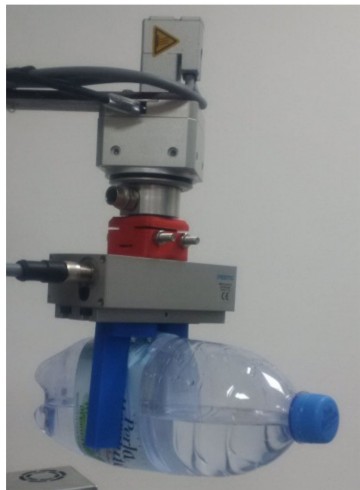

**Figure 6.** The gripper mechanism in action.

Once the gripper is attached, we can calculate the required force needed to grab certain recyclable materials/objects. To achieve this, we can use the stick–slip conditions [42] derived from the Coulomb friction law [43]. Figure 7 presents the forces acting between the gripper's fingers and the object being manipulated, where:

$\overrightarrow{F}_{f_L}$ = Friction force between left finger and the object

$\overrightarrow{F}_{f_R}$ = Friction force between right finger and the object

$\overrightarrow{F}_{C_L}$ = Control force on the left finger

$\overrightarrow{F}_{C_R}$ = Control force on the right finger

$\overrightarrow{G}$ = Object's weight

$\overrightarrow{N}_L$ = Normal force on left fingers' contact surface

$\overrightarrow{N}_R$ = Normal force on right fingers' contact surface

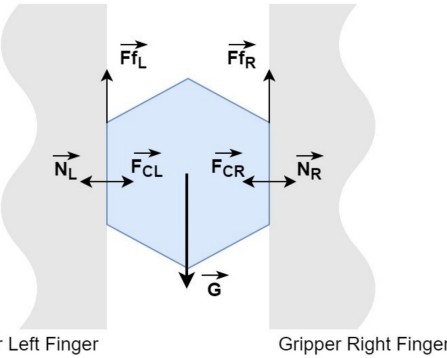

**Figure 7.** Forces acting on gripper fingers and the grabbed object.

Coulomb friction law states that for the object to not slide from the gripper fingers, the forces must follow Inequality (1), where the friction forces are defined in Equations (2) and (3):

$$\vec{F}_{f_L} + \vec{F}_{f_R} \geq \vec{G} \tag{1}$$

$$\vec{F}_{f_L} = \mu \vec{N}_L \tag{2}$$

$$\vec{F}_{f_R} = \mu \vec{N}_R \tag{3}$$

In our case, the control forces $\vec{F}_{C_L}$ and $\vec{F}_{C_R}$ are equal with the gripper's control force $\vec{F}_C$, resulting Equations (4) and (5):

$$\vec{F}_{C_L} = \vec{F}_{C_R} = \vec{F}_C, \tag{4}$$

$$\left. \begin{array}{l} \vec{N}_L = \vec{F}_{C_L} \\ \vec{N}_R = \vec{F}_{C_R} \end{array} \right\} => \left\{ \begin{array}{l} \vec{F}_{f_L} = \mu \vec{F}_{C_L} = \mu \vec{F}_C \\ \vec{F}_{f_R} = \mu \vec{F}_{C_R} = \mu \vec{F}_C \end{array} \right\}. \tag{5}$$

From Equations (1)–(5) we calculate that the required condition for the gripper to hold the recyclable object is the one from the relation:

$$2\mu \vec{F}_C \geq \vec{G} \tag{6}$$

Using this inequality, we can further analyze the required force depending on the estimated weight of the object and the friction coefficient for certain types of recyclable materials. However, before analyzing the minimum required control force, we have to take into consideration the gripper motion on the same axis as the friction forces. Since the gripper will move up after the object is grabbed, the dynamic motion will add a new acceleration to the object, which will, in turn, translate into an inertial force that has the same direction as the weight of the object. This means that we need to increase the force acting on the object. Equations (7)–(9) add the new inertial force to the control force condition from Equation (6).

The maximum acceleration for the motor placed on the vertical axis, as given by the manufacturer [44] is:

$$a_i = 10\text{m/s}^2. \tag{7}$$

Then, the inertial force added to the weight is:

$$\vec{F}_i = m \times a_i. \tag{8}$$

From Relations (1) and (8) we obtain:

$$2\mu \vec{F}_C \geq \vec{G} + \vec{F}_i. \tag{9}$$

Expanding this relation, we get:

$$2\mu \vec{F}_C \geq m(g + a_i), \tag{10}$$

where:
  $\vec{F}_i$ = Inertial force due to vertical motion
  $a_i$ = inertial acceleration due to vertical motion
  $m$ = Objects' mass
  $g = 9.81\text{m/s}^2$ = Gravitational acceleration
  $\mu$ = Friction coefficient

Knowing the maximum acceleration (Equation (11)) and the maximum force [45] (Equation (12)) that the gripper can sustain, we can compute the maximum object mass that our gripper can hold for certain materials with known friction coefficients:

$$a_{max} = g + a_i = 19.81 \text{ m/s}^2 \tag{11}$$

$$\overrightarrow{F}_{C_{max}} = 250 \text{ N}. \tag{12}$$

From Relations (10) and (12) we can compute the required force for grabbing an object and we can compare it with the maximum force the gripper can apply:

$$\overrightarrow{F}_C \geq \frac{m(g + a_i)}{2\mu}. \tag{13}$$

If we consider the recommendations presented by online references [46,47], the control force should be four times higher than the required force for gripping the object, without considering the vertical acceleration, and is given by the relation:

$$\overrightarrow{F}_C = \frac{mg}{2\mu} \times 4. \tag{14}$$

**Table 2.** Static friction coefficients.

| Material | Friction Coefficient |
| --- | --- |
| ABS—PET | 0.33 [48] |
| ABS—Glass | 0.3 [49] |
| ABS—Aluminum | 0.25 [49] |

This means that we can add a safety margin and all objects regardless of the systems' accelerations will be firmly held by the gripping system. However, since Relation (13) takes into account the maximum acceleration of the vertical axis, and we do not expect any other forces to influence the system, we will use Relation (13) as the reference for computing the force needed by the system to pick up objects with different friction coefficients and weight. While the maximum acceleration is given by Relation (11), we can relate to Relation (14) and present the new force control as:

$$\overrightarrow{F}_C = \frac{mg}{2\mu} \times 2, \tag{15}$$

where the new safety margin is 2. However, in real experiments, we will use Relation (13) as it is more accurate.

In our case, the recyclable materials of interest are plastic bottles, glass bottles, and aluminum cans. Knowing this, we will continue the analysis using these materials. Thus, Table 2 presents the known static friction coefficients between the ABS fingers and PET, glass and aluminum objects, where we consider them more dry than wet.

Once we have the maximum force relation and the friction coefficients, we can compute the required force for the gripper to pick up a certain object. Figure 5 presents the computed force control value for different object mass and material types. The figure shows the maximum force limit, represented by the top dashed rectangle. Using this diagram (Figure 8), we can visualize the maximum weight of the recyclable objects that the gripper can pick up and transfer to the recycling tray. Since for all three materials the maximum weight is above 6 kg, we can presume that all recyclable objects that fit the recycling criteria can be grabbed by our 5DOF gripper.

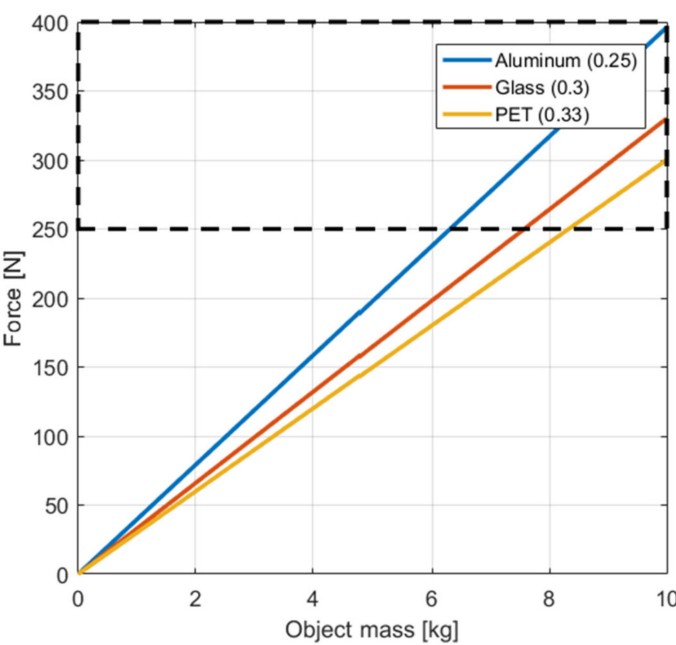

**Figure 8.** Gripper force by object mass and friction coefficient.

By using Equation (13) and the friction coefficients of the recyclable materials of interest (Table 2) we can now use this information in the hybrid position/force control architecture to compute the gripper reference force.

## 5. The 5DOF Hybrid Position/Force Control

To achieve a functional control method, the mobile robots' manipulator, used for sorting waste materials, has to be able to move at target points and then grab the recycling materials using a simple gripper. For this, the hybrid position/force control method [15] was chosen and then modified to integrate the systems' state, changing the control type of the manipulator when needed, between position and force control mode. Thus, we changed the static selection of control type for each degree of freedom to a deterministic state of the S-matrix that separates each DOF into two categories: control mode and force mode. Figure 9 presents the new hybrid position/force control architecture used to drive the 5DOF manipulator.

Figure 9 presents the data flow of information between each control block, where:
$O(p, type)$ = Object (position, type);
type = Glass, Aluminum, Plastic;
$O(depth)$ = Object (depth);
Depth = distance from the robot reference system to the detected object.

In the new architecture of the hybrid position/force control method of Figure 9, we have included several control blocks that group the control architecture by task, while the arrows present the data flow. The control diagram starts with the reference generation block. This block receives information from the video detection system that will provide the objects' position in XOY coordinates and the object type. The type is then used by the reference generation to compute the required force according to the friction force coefficient influence. Another input is received from the proximity sensors, which will provide the distance to the target object, which represents the third coordinate in the 3D space, the OZ axis.

The video detection system [50] uses neural networks to detect the type, position, and orientation of the recyclable objects and will send this information as input for the 5DOF manipulator, through a TCP network interface. Combining both video detection for XOY coordinates and orientation with the proximity sensors for depth distance, we get the OXYZ coordinates with θ angle orientation. The detection system was developed by

our project team and is already published by Melinte et al. [50], which is why we will not focus on its results in this paper, but we are using the AI detection module as input to our control architecture. The video camera used has an RGB resolution of 1280 × 1024 pixels, which is more than enough for our AI detection system to use, with a detection confidence of 75.54% and over 95% accuracy [50], using a 9 FPS image analysis rate.

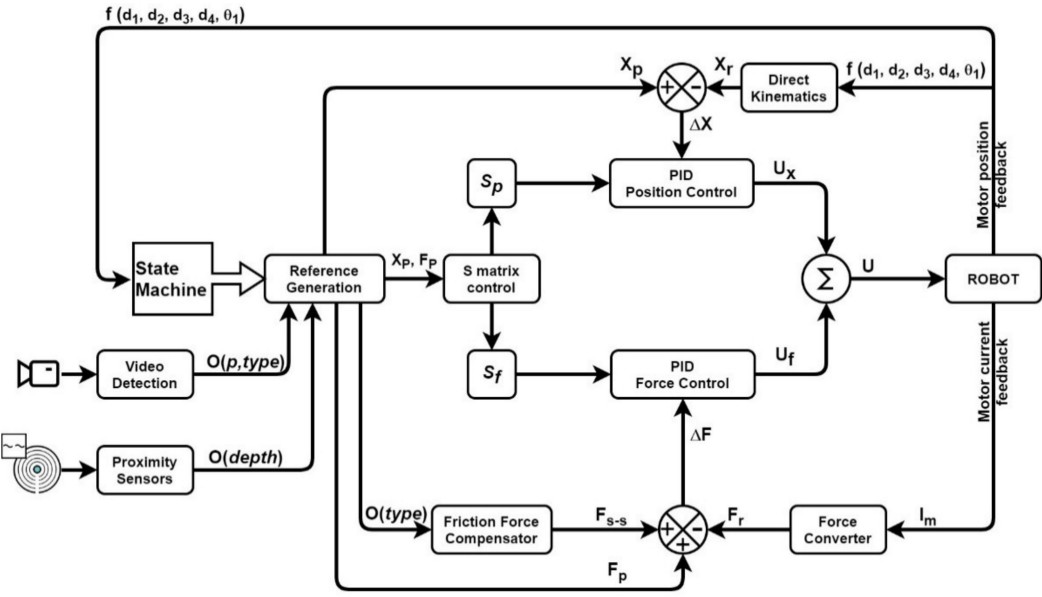

**Figure 9.** Hybrid position/force control diagram.

Added to the input data, the systems' state, provided by the state machine, the robot can now decide what to do next, to achieve the given recycling task. If the robot is in search mode, the manipulator is resting at the starting position. Then, when an object is detected, the systems' state switches to the start state of the pick-and-drop process. When the state of the system requires the manipulators' gripper to switch in order to force control then the "S-matrix control" block will update the S-matrix, by switching the control type of the gripper DoF. Equation (16) presents the way the S-matrix is composed out of the $S_p$ and $S_f$ matrix:

$$S = \begin{bmatrix} 1 & \cdots & 0 \\ \vdots & \ddots & \vdots \\ 0 & \cdots & 1 \end{bmatrix} = S_p + S_f = \begin{bmatrix} 1 & . & . & . & 0 \\ . & 1 & & & . \\ . & & 1 & & . \\ . & & & 1 & . \\ 0 & . & . & . & 0/1 \end{bmatrix} + \begin{bmatrix} 0 & . & . & . & 0 \\ . & 0 & & & . \\ . & & 0 & & . \\ . & & & 0 & . \\ 0 & . & . & . & 1/0 \end{bmatrix}. \quad (16)$$

After the reference values are calculated, the hybrid method splits into two control branches, each dealing with a specific control type: position control and force control. Each control block uses a PID to compute the joint-specific drive inputs (Equations (17)–(19)):

$$U_x = X_{PID}(S_p, \Delta X), \quad (17)$$

$$U_f = F_{PID}(S_f, \Delta F, F_{S-S}), \quad (18)$$

$$U = U_x + U_f. \quad (19)$$

One can see that while the position PID regulator depends on the positioning error between the reference and the computed value, using the direct kinematics method and Denavit-Hartenberg notation [51], the force PID regulator will have as input the force control error computed using the reference value, the real force taken as a function of motor

current and the force required to compensate the object weight through the friction force compensator control block.

Equation (20) presents the function that receives the robot joint information and sends it to the state machine and direct kinematics blocks:

$$f(d_1, d_2, d_3, d_4, \theta_1). \tag{20}$$

Using the joint information, Equations (21)–(23) present the elements required to compute the positioning and force errors, which are inputs to PID regulators that, in the end, drive the 5DOF manipulators' motors.

$$U = X_{PID}(S_p, \Delta X) + F_{PID}\left(S_f, \Delta F\right), \tag{21}$$

where:

$$\Delta X = X_P(State, Obj(p, type), Obj(depth)) - X_r(Rob_{kin}(d_1, d_2, d_3, d_4, \theta_1)) \tag{22}$$

$$\Delta F = F_P(State) + F_{s-s}(Obj(type)) - F_r(F_{conv}(I_m)) \tag{23}$$

To obtain the PID position control values, we used the direct kinematics equations through Denavit-Hartenberg notation that are shown in Figure 10 and presented in Equation (24).

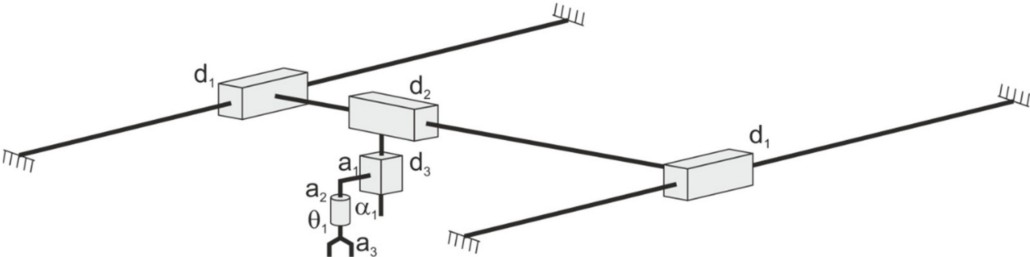

**Figure 10.** Kinematics diagram and D-H notations of the 5DOF, TTTRT manipulator.

Since $\alpha$ is $0°$, then the direct kinematic coordinates are given by Relation (24), where $a_1$ and $a_2$ are constants given by the mechanical structure dimensions. The gripper's degree of freedom is not taken into account when computing the direct kinematics matrix. This is because we had to position the gripper near the desired object, and then close the gripper's jaws (5th degree of freedom).

Using the direct kinematics matrix:

$$\begin{bmatrix} cos\theta_1 & -sin\theta_1 & 0 & -a_1 + d_1 \\ sin\theta_1 & cos\theta_1 & 0 & d_2 \\ 0 & 0 & 1 & -a_2 - d_3 \\ 0 & 0 & 0 & 1 \end{bmatrix}, \tag{24}$$

and the PID general control equation:

$$PID = K_P e(t) + K_i \int_0^t e(t)dt + K_d \frac{de(t)}{dt}, \tag{25}$$

the position and force controllers were computed following classic methods [52]. Since the PID controllers are not in the scope of this paper, we will not present their design and tuning procedure.

The hybrid method we have proposed in this paper uses hardware components that have a clear separation between position and force control. Because of this, the stability problem is avoided, since when switching the control method, the regulator method is

changed for that specific motor (DoF), and the motor is in a paused state (not moving). When changing the control method, we can say that the control mission is restarted, which is why there is no stability issue at the switching moment within the control law.

## 6. Experiment

The hybrid position/force control is designed for the 5DOF manipulator which will be mounted on the mobile robot. The manipulator has several hardware components that build the positioning and orientation system and one linear motor for the end effector.

Figure 11 presents the interaction diagram of the hardware components. One can see that the system has only one PLC system that sends the reference information to the planar XOY movement (Festo EXCM), to the vertical OZ positioning (Festo EXCM), and the rotary motor (Festo ERMO), through CANopen, Modbus TCP or Input/Output Link (IO-Link) communication with integrated motor drivers. The gripper's motor requires a separate motor driver (Festo CCEC-X-M1) that controls the linear motor (Festo HGPLE), which forms the gripper with the attachment of two 3D printed ABS fingers.

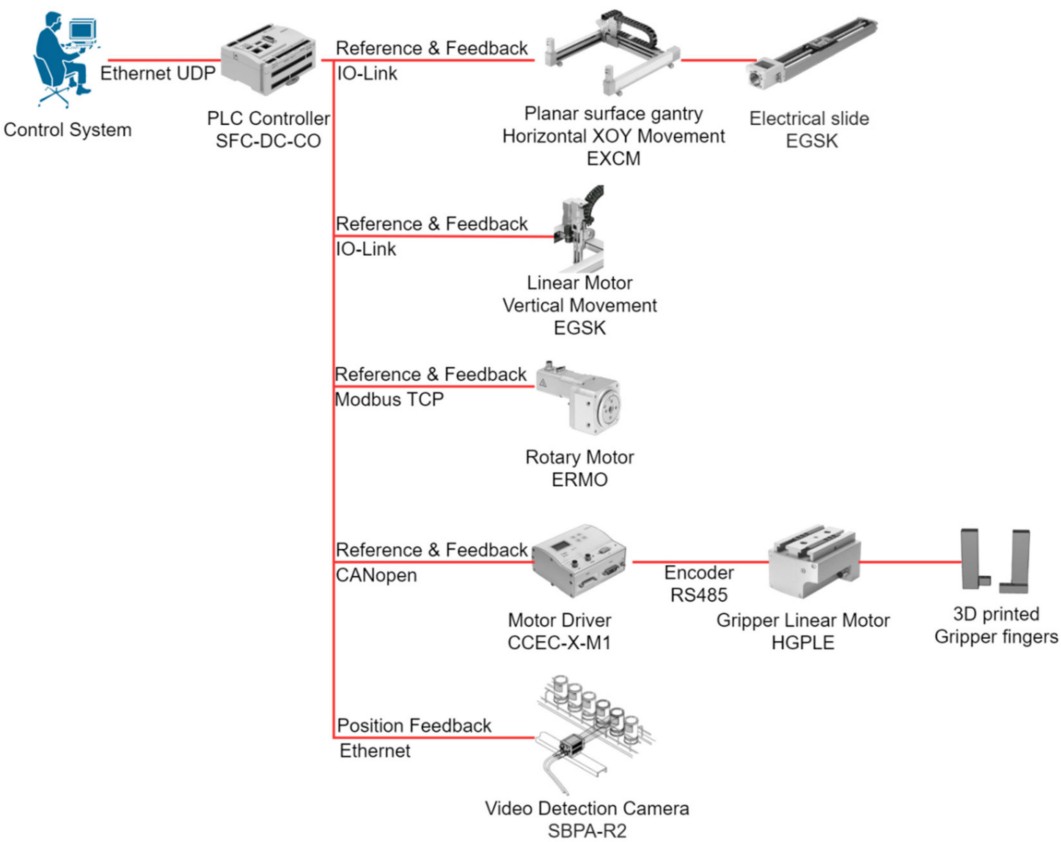

**Figure 11.** The robot control components interaction diagram.

The experiment uses a video detection camera, which uses a neural network [50,53] to detect different recyclable materials (glass bottles, aluminum cans, PET bottles) and sends their position and orientation to the 5DOF manipulator as input information. This information was simulated in the first stage of the experiment, since we had to prove that the hybrid position/force control has a good performance in the process of controlling the manipulator throughout the recycling operation.

Because the gripper's control was the one that will switch between position and force control, we can say that this joint (DoF) is the most important one when deciding if the hybrid control can be used for this particular task, and we will describe its process in more detail.

The 5DOF manipulator is controlled by a Festo PLC, programmed using the CoDeSys software. Using the state machine presented in Figures 3 and 4, one can see that the entire system has several control steps. The first stage is to initialize the motors and start the homing process for each motor, and then wait for the detection system to send the first signal of position, orientation, and type for the object to recycling. After the initialization is complete, the recycling system can receive the pick-and-drop commands.

To control the 5DOF mechanical system, a decision algorithm that implements the state machine was developed outside the PLC, with communication capabilities through Ethernet-UDP protocol (Figure 12). This system receives state messages from the manipulator and the video detection system, and, through a transition mechanism, sends appropriate commands to the manipulator. The messages have two parts: command name and value. The message is then decoded by the PLC through a decoding method. The decoded information is used to set motor parameters and then activate the command. While the motor is acting on the last received command, it will send status information through a transducer to the motor driver which, in turn, is received by the PLC that will send status information through Ethernet UDP to the state machine.

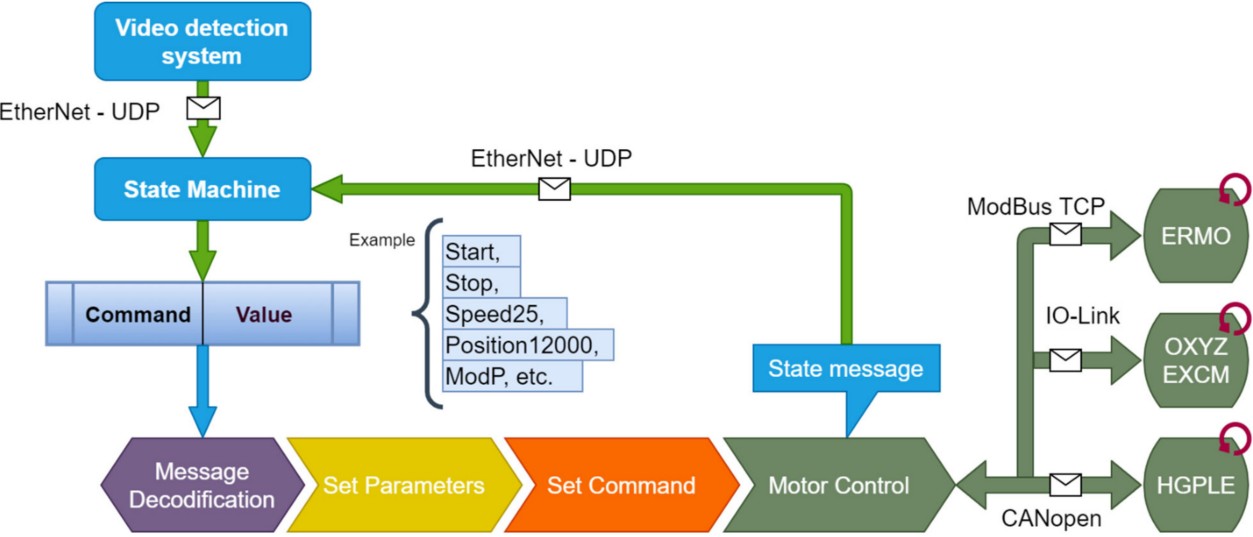

**Figure 12.** Motor control through PLC programming and state machine commands.

To test the gripper's commands and states, we designed a CoDeSys visualization interface that shows the state of each command or status bit.

Figure 14 presents all the main stages of the gripper's control. The left part of each figure contains the control bits, and the right part contains the status bits. As the name suggests, the control bits are the values sent to the motor driver and the status bits are the values received. Grey status means that the bit is not used by the driver, dark green means that the bit value is 0 (false) and light green means that the bit value is 1 (true). Using these control bits, we can set the motor driver states from initialization and homing to the start of positioning or force control. Thus, Figure 14a,b present the configuration bits for the initialization and homing functions of the motor driver, respectively. After the initial step and homing have been completed, the motor driver, and consequently the HGPLE motor, are in stand-by mode to receive control values. The first values that will be sent to the driver are the control mode: position control mode (Figure 14c) or force control mode (Figure 14e). After these values comes the speed reference with which the gripper will move, while trying to reach its position or force reference. The third step is to send the position and force reference values and to start the control process. Figure 14d shows position control and Figure 14f shows force control. While the positioning mode will stop after the position is reached, the force control will continuously try to hold the set force.

The force control can be stopped by setting the control force to 0, by manually stopping the control process, or by switching the control type to positioning mode.

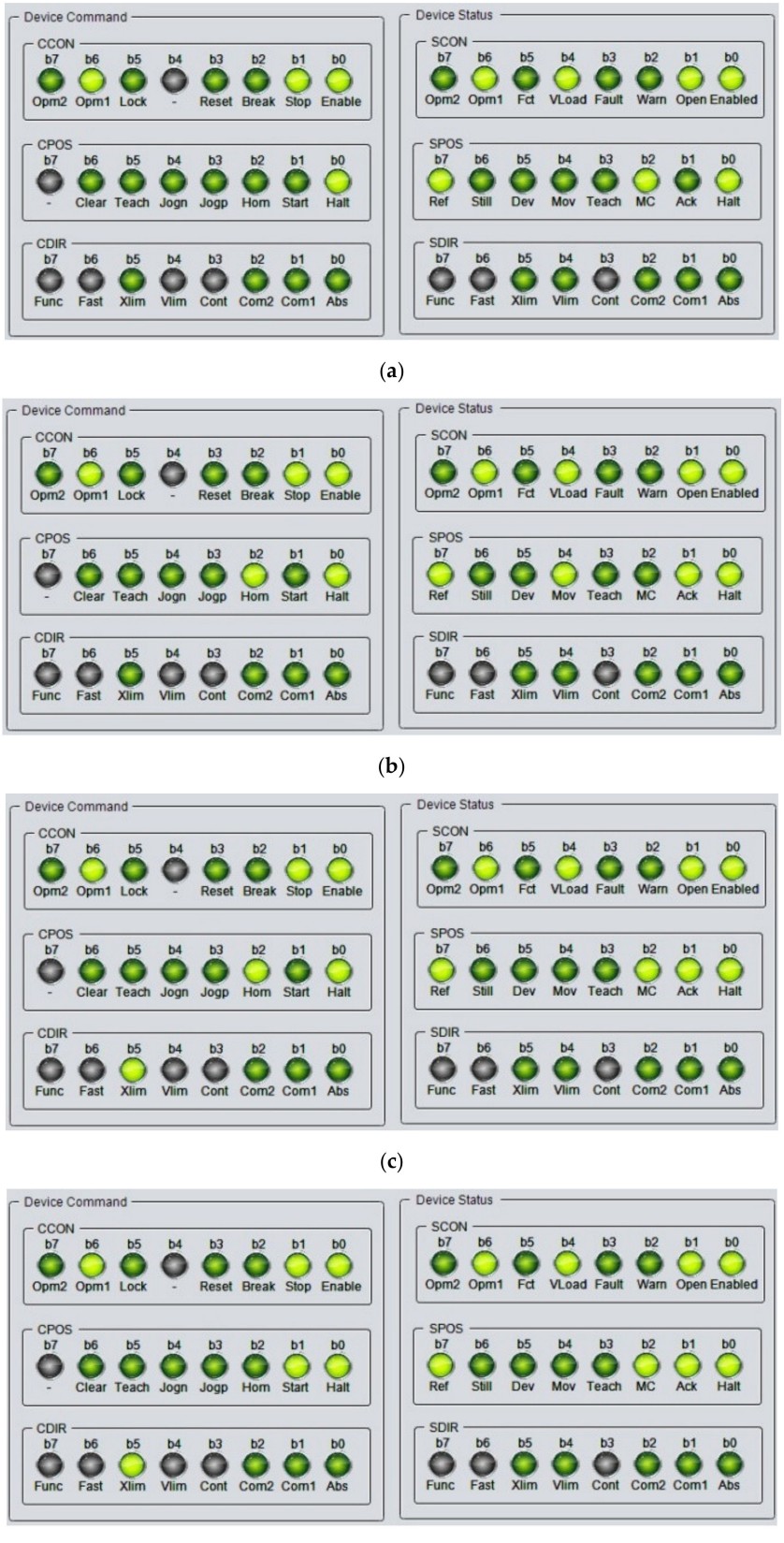

**Figure 13.** *Cont.*

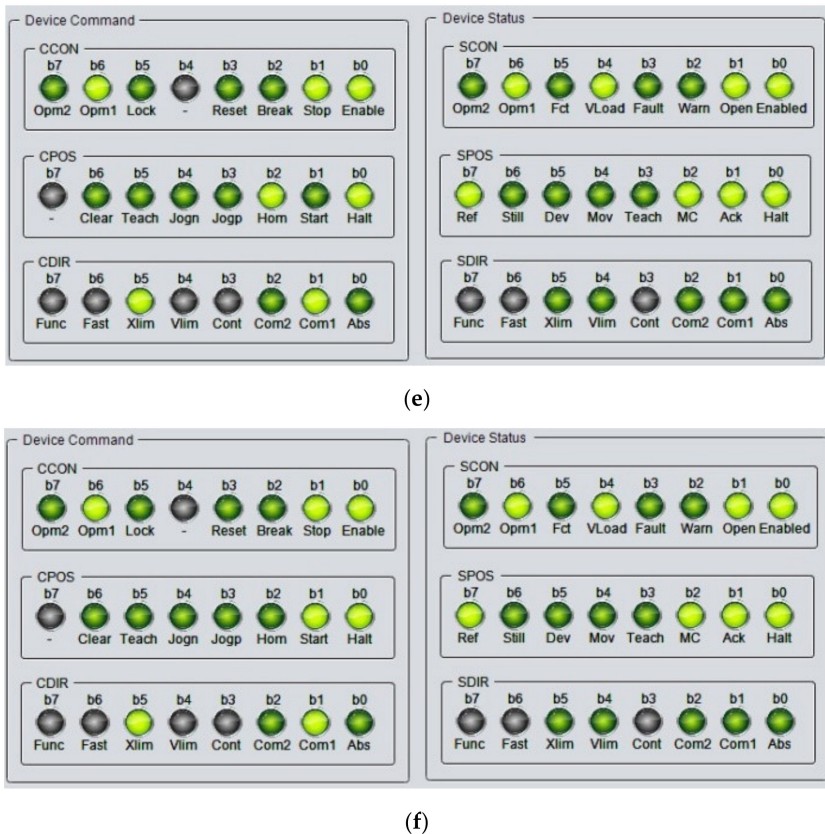

(e)

(f)

**Figure 14.** CoDeSys visualization interface: (**a**) initialization stage; (**b**) homing stage; (**c**) positioning mode control; (**d**) start positioning control; (**e**) force mode control; (**f**) start force control.

Using the described steps, the gripper can be used in the hybrid position/force control method for the 5DOF manipulator. While the positioning and orientation system will always be controlled in position, the gripper will switch between position and force control states depending on the system's state machine.

Figures 15 and 16 present the control phase for position control and force control, respectively, for the manipulator's gripper.

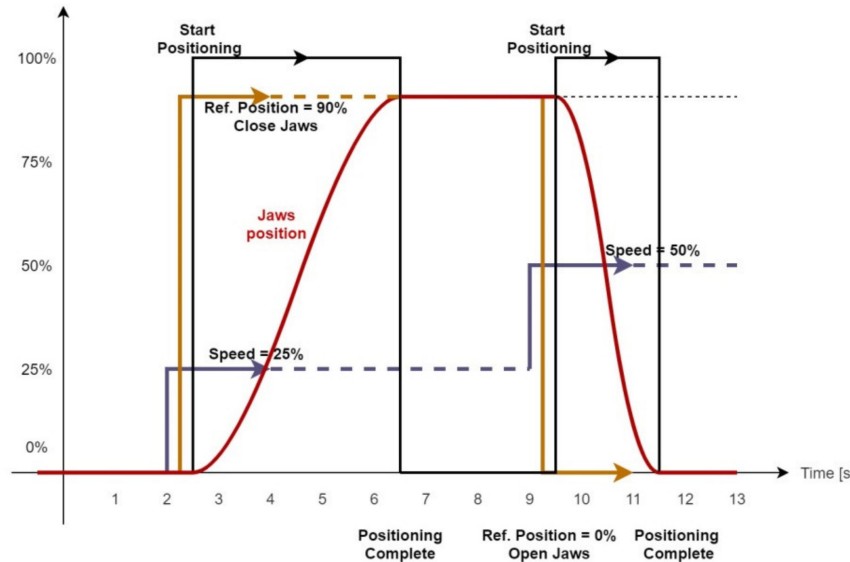

**Figure 15.** Gripper position control states.

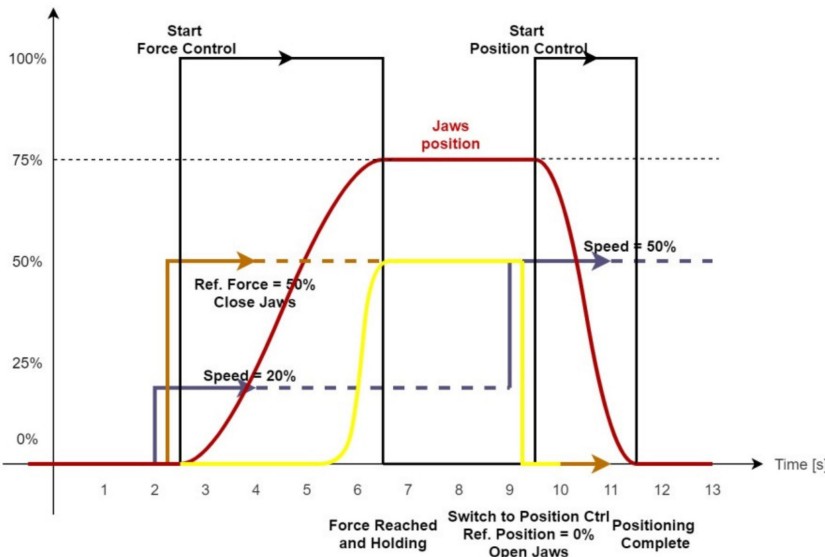

**Figure 16.** Gripper force control states.

The color code used in Figures 15 and 16 is:

- Purple line = speed reference;
- Brown line = force and/or position reference;
- Black line = start command and auto-stop state;
- Red line = gripper jaws position;
- Yellow line = gripper applied force.

For positioning control, Figure 15 is quite self-explanatory, where the reference velocity is the first to be set, followed by the position reference. Once these two have been set, the start command can be given, and the positioning regulator will start moving the motor to the desired position with the configured speed. The gripper jaws will stop when the target position is reached or when an alarm is detected. The alarm can be triggered by an overcurrent due to an obstacle within the jaws' path. When the position is set to 0 to open the gripper jaws, and the start command is given once again, the gripper jaws will move with the newly set speed.

For the force control, the control diagram is more complex and is presented in Figure 16. This control process starts similarly to the positioning process, with setting the speed reference. The next step is to set the target force and to start the control process. Before the target force is reached, the gripper will move until an object/obstacle is reached and the control force will increase. When the reference force is reached, the regulator will not stop and will continue to feed current to the motor. At this point, the manipulator's 4DOF positioning and orientation system can move the recyclable object. When the target OXYZ position is reached, the gripper's state is switched from force to position control, set the reference position to 0, and start the positioning control to open the gripper's jaws.

Using the described process of the hybrid position/force control to pick-and-drop recyclable waste, we tested it for several objects and materials. Figure 17a–c present three of the objects being grabbed and held until the movement state is complete and the gripper can drop the recyclable material. As one can see, Figure 17a–c present the orientation and gripper components. We tested the two degrees of freedom more thoroughly than the entire 5DOF system, since, for these two, the hybrid position/force control is more important.

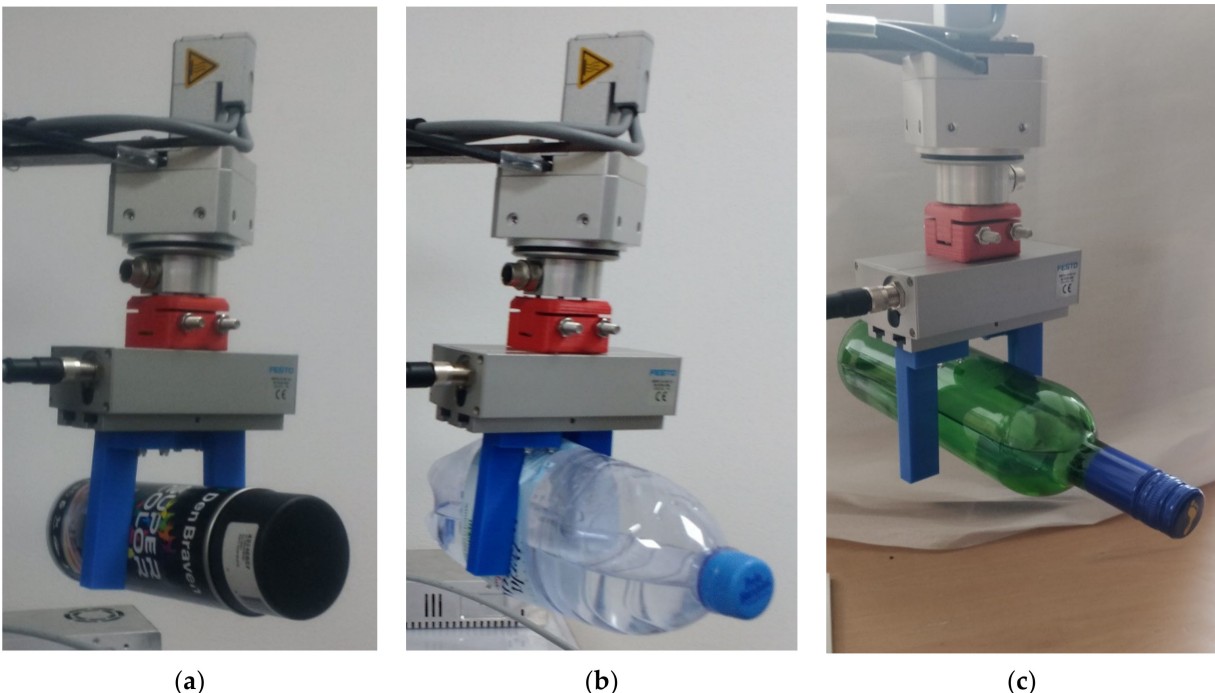

**Figure 17.** Experiments on object gripping: (**a**) experiment for aluminum pickup; (**b**) experiment for PET bottle pickup; (**c**) experiment for glass bottle pickup.

## 7. Discussion

In order to use a mobile robot with a 5DOF manipulator for picking up recyclable material, the first stage is to initialize the robot's components. This means that for each motor and actuator, a homing process is required when the robot starts up.

After the first initialization stage is complete, the robot is ready to start its search-and-recycle task. For that, a human operator starts the mission, and the robot will move within the mission area to search for recyclable materials, using its neural network-driven detection video camera. At this stage, the manipulator is resting in the homing-done-ready-for-task state.

When the robot detects a target object, it will compute the next stage reference for the position, orientation, and type of the object to pick up. At this moment, the state of the system's manipulator changes from resting to start-task. All reference values are, at this stage, being used to provide accurate task information for the manipulator. This data includes a position in 3D space, orientation of the target object, and estimated friction between the gripper's fingers and the target. Using these values, the robot will start its mission with a positioning task, followed by a hybrid position/force control to grab the object and place it in the recycling tray. The last task is to position the manipulator in a starting position with the gripper jaws open.

As seen from the main stages of the pick-and-drop process, the proposed hybrid position/force control architecture based on a state machine is very useful for this type of robot and mission.

For hybrid methods that require an online switch mechanism, the interest is quite high between researchers and manufacturers. However, only certain types of motors can be controlled in this way. An example is the research of Pasolli and Ruderman [38,54], which proposed a hybrid control to be used on actuators which may not require different control parameters, and should be used where the actuators' parameters rarely change, while the control values can be modeled as a single transfer function, within the control architecture diagram.

The main difference between our approach and the one used by Pasolli and Ruderman [38,54] is that our hardware has parameters that require an update every time the

control mode is switched. This is because modern motors and actuators are produced with built-in regulators or fail-safe mechanisms to protect the hardware and provide the engineer with a basic motor controller. With these types of motors and actuators, you need a different type of hybrid control, one that can update the parameters of the motor while switching the control type with certain degrees of freedom, and one that can use different control methods.

Similar hybrid control with a built-in switching mechanism developed by Pasolli and Ruderman [38,54] is used by the majority of hybrid control methods [16,18,19,24,26–30,33–35,38,54,55], where the motor allows position or force control just by increasing the driving current or motor torque. These are better used with a dynamic control law integrated into the hybrid position/force control method. The built-in switching mechanism that allows the hybrid control to control a certain motor either in position or in force requires a stability analysis, but when the motor has to switch parameters to commute between position and force control, the stability analysis is not needed. Since specific motors require new parameters to switch between control types, kinematic control methods are enough to drive the robot joints.

The hybrid position/force control with an online switch is not unfamiliar to us [20], but it was considered to be unnecessary for the task at hand. We have developed other control methods that require stability analysis [20], and ones that use dynamic control methods [20,25], along with the hybrid position/force control to drive robots in their mission. Compared to our previous research, the approach presented in this paper makes it easier to implement the control method, but it has a higher complexity when using these motors. The complexity comes from the multitude of parameters required to adjust not only when commissioning the motor, but also at run time when switching the required parameters to drive the motor in either position or force/torque.

Knowing this, the proposed architecture of the state machine-driven hybrid position/force control is useful for many other robots [55,56] that are being used in different environments and for different tasks, but more importantly, for those that are being used in automated processes within Industry 4.0 type factories. These factories rely on the fact that the robots can switch tasks whenever the supervisor asks them to change the production line. Since our proposed architecture is based on a state machine, one can design a way to update the state machine for each type of mission and task the robot is given, gaining not only a hybrid position/force control method but also an adaptable robot with a set of missions to choose from.

## 8. Conclusions

The proposed architecture of the state machine-based hybrid position/force control (SmHPFC) can be implemented for robots and actuator-driven mechatronics, which use motors with different control parameters between position and force control. One example is the HGPLE Festo linear motor. This type of motor tends to include feedback information on the real position and force/torque that can be easily included in a feedback control loop such as the SmHPFC. Using this information, one can design a position control feedback loop, knowing that the received information is showing the actual actuator position. At the moment, when the control type is changed from position to force or torque and vice versa, the same information value must be interpreted with a different scope, which will disturb the real-time control loop since the motor control parameter has changed. In the standard approach, a motor is usually controlled in position, force/torque, or direct current, and uses different techniques to get the control value, starting from a position or force/torque reference. This approach cannot be used by a state-driven actuator such as the one used on the presented 5DOF manipulator.

One advantage of the SmHPFC architecture is that it is quite easy to integrate it into an automated assembly line or use it to change the behavior of the robot since the stability problem was mitigated due to the way the motors are built. This advantage translates

into reconfiguring the automated line with ease, which is the backbone of the Industry 4.0 concept.

As an example of Industry 4.0 usage, the robots from an assembly line can be programmed to complete different tasks, while in a standard production line, the extra programmed tasks are not required. A robot can have a task to move certain components from a top shelf to the conveyor in a certain factory configuration, but can also stack products to a designated area, depending on the factory requirements. This can happen if all the robots are pre-programmed with multiple tasks and a control unit will change the robot state from a shelf-stacking robot to one that can pick up faulty products from the conveyor. Another advantage of the SmHPFC architecture is that not only can it provide easy access to multiple types of tasks for a single robot, but that these tasks can have very different control parameters and objectives: position control, contour following, force control, etc.

This type of architecture has some disadvantages. The main one is that it cannot switch between control modes while the robot is moving and it must perform a very short stop while reconfiguring the input parameters. This stage is dependent on the communication rate between the PLC and the motor, and on the time the motor can come to a stop. It takes only a few communicated messages for the motor to stop, reconfigure and start a new job. For example, to switch the Festo linear motor from position to force, we need to send just one control byte. However, since the motor parameters cannot be changed while it runs, we have to send two additional control bytes, to stop and restart the motor. On a ModBus 9600 bps communication rate, it would translate to 2.5 ms, only to send the new configuration. To this, we need to add the time it takes the motor to come to a stop if not already stopped, and a maximum of eight more bytes to send the new motor reference. This also depends on whether the motor can restart with ease. This disadvantage can turn into an advantage if the robot task requires separate motion steps during which the robot can be reconfigured to handle different control methods on a different degree of freedom.

Our proposed SmHPFC architecture does not depend on the joint regulators, and their analysis was not included in this paper. Thus, the presented applied research is meant to prove the behavior of our architecture and provide new means of control for automated Industry 4.0 lines, but also for motors that are not as easy to integrate into a fully automated robot due to restrictions in their control and parameterization functions.

Future work will focus on integrating the deterministic artificial intelligence [57] that provides clear analytic methods for reference trajectory generation on each type of control. By improving the reference aspect, we expect to improve the overall architecture, transforming the reference generation component into a more robust and reliable component that can provide the best data for the robot's current task and control type.

**Author Contributions:** Control architecture, I.-A.G.; conceptualization, I.-A.G.; control algorithm, I.-A.G., formal analysis, I.-A.G., A.-C.C. and M.M.; resources, I.-A.G., A.-C.C. and M.M.; writing—original draft preparation, I.-A.G. and A.-C.C.; writing—review and editing, I.-A.G.; funding acquisition, I.-A.G. and M.M. All authors have read and agreed to the published version of the manuscript.

**Funding:** This work was supported by a grant from the Romanian Ministry of Research and Innovation, CCCDI—UEFISCDI, project number PN-III-P1.2-PCCDI-2017-0086/contract no. 22 PCCDI/2018, within PNCDI III, and with the support of the Robotics and Mechatronics Department, Institute of Solid Mechanics of the Romanian Academy.

**Institutional Review Board Statement:** Not applicable.

**Informed Consent Statement:** Not applicable.

**Data Availability Statement:** Not applicable.

**Acknowledgments:** This work was supported by a grant from the Romanian Ministry of Research and Innovation, CCCDI-UEFISCDI, MultiMonD2 project number PN-III-P1.2-PCCDI2017-0637/33PCCDI/01.03.2018, within PNCDI III, and by the European Commission Marie Sklodowska-Curie SMOOTH project, Smart Robots for Fire-Fighting, H2020-MSCA-RISE-2016-734875 and by inter-academic project IMSAR–Yanshan University: "Joint Laboratory of Intelligent Rehabilitation

Robot", KY201501009, collaborative research agreement between Yanshan University, China and Romanian Academy by IMSAR, RO. The authors thank S.C. Festo S.R.L., Bucharest, Romania, for their technical guidance and assistance in using the Festo products concerned in this paper.

**Conflicts of Interest:** The authors declare no conflict of interest. The funders had no role in the design of the study; in the collection, analyses, or interpretation of data; in the writing of the manuscript, or in the decision to publish the results.

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
