# Peer review of "State Machine-Based Hybrid Position/Force Control Architecture for a Waste Management Mobile Robot with 5DOF Manipulator"

_applsci, doi:10.3390/app11094222_

Round 1
Reviewer 1 Report
- The abstract is weak, since no results are described. “very useful for the Industry 4.0 automation lines” is insufficient to entice readers to read the manuscript to learn its contents. Please add some quantitative results to the abstract and conclusions.
- Citation is poorly done in section 1 with many multiple citations with no clear rationale where it seems a single citation is appropriate to reduce the burden placed on the reader. Subsequent citations in sections are well done.
- Please add specific details why readers should seek each of the triple citations [1-3].
- Please add specific details why readers should seek each of the double citations [5,6].
- Please add specific details why readers should seek each of the double citations [7,8].
- Please add specific details why readers should seek each of the triple citations [8-10].
- Please add specific details why readers should seek each of the double citations [13,14].
- Please add specific details why readers should seek each of the double citations [21,23].
- Please add specific details why readers should seek each of the triple citations [28-30].
- Please add specific details why readers should seek each of the double citations [34-35].
- Figures are generally well done.
- The text inside figure 11 is far too small to be discernable in printed copies of the manuscript.
- Figure 12 is marginal, but acceptable.
- Figure 13 is rendered useless, since the text inside the font is so small as to relay no information to the reader.
- The strength of the conclusion is its confession of disadvantages bestowing believability to the claims made in the manuscript. There is no hint of future research. The reviewer would very much like to see application of motor control using deterministic artificial intelligence (recently published for underactuated cases) to see if the proposal here in this submitted manuscript to learn if augmenting updates with switching could amplify the results achieved by D.A.I.
Reviewer 2 Report
In my view, the article has two distinct parts. In introduction waste management but in the experiment only hybrid position/force control without follow-up to waste management. An adequate introduction in which the problem under study is focused and adequately explained, and a development that needs to be improved in depth. I would suggest that the authors consider these recommendations:
- The resolution and quality of some figures (e.g., Figure 1 and Figure 13) should be improved. In Figure 13, I would consider its contribution to the article.
- In the experiment is for the position feedback used industrial camera Festo SBPA-R2. Why it was chosen? Is fully sufficient for the given purposes? Wouldn't it be better to choose a solution built specifically for the purpose with 3D vision cameras (scanners) and AI? I miss the justification for the choice in the article.
- The robot will collaborate with the environment. How is its protection against an unwanted collision solved? How the robot's surrounding work environment is monitored? The article lacks a justification for securing a mobile robot when manipulating the arm.
- The camera uses external object recognition software. What is its processing speed? It is sufficient for real-time operations? The article lacks basic information about system performance. It would be appropriate to add them there in order to assess the benefits of the solution for real practice.
- Figure 16 lacks a picture of gripping a glass bottle, why? How is the magnitude of the pressing force solved for different materials?
- I miss the description in the article on the basis of which the objects were recognized. What algorithm and rules were chosen to distinguish waste from common objects?
- What do the authors see as their contribution and novelty, unlike other solutions? In the Discussion chapter, I miss the comparison with alternative concepts and highlighting the benefits of the approach described in the Article.
- In the references, the authors mentioned articles by the author Vladareanu six times, it is really necessary?
Round 2
Reviewer 2 Report
All comments and recommendations have been incorporated.I have no further reservationsabout the article.